# High-Precision Tribometer for Studies of Adhesive Contacts

**DOI:** 10.3390/s23010456

**Published:** 2023-01-01

**Authors:** Iakov A. Lyashenko, Valentin L. Popov, Roman Pohrt, Vadym Borysiuk

**Affiliations:** 1Department of System Dynamics and Friction Physics, Institute of Mechanics, Technische Universität Berlin, 10623 Berlin, Germany; 2Department of Applied Mathematics and Complex Systems Modeling, Faculty of Electronics and Information Technology, Sumy State University, 40007 Sumy, Ukraine; 3Department of Nanoelectronics and Surface Modification, Faculty of Electronics and Information Technology, Sumy State University, 40007 Sumy, Ukraine

**Keywords:** force sensor, tribometer, drift, hysteresis, contact mechanics, contact area, adhesion

## Abstract

Herein, we describe the design of a laboratory setup operating as a high-precision tribometer. The whole design procedure is presented, starting with a concept, followed by the creation of an exact 3D model and final assembly of all functional parts. The functional idea of the setup is based on a previously designed device that was used to perform more simple tasks. A series of experiments revealed certain disadvantages of the initial setup, for which pertinent solutions were found and implemented. Processing and correction of the data obtained from the device are demonstrated with an example involving backlash and signal drift errors. Correction of both linear and non-linear signal drift errors is considered. We also show that, depending on the research interests, the developed equipment can be further modified by alternating its peripheral parts without changing the main frame of the device.

## 1. Introduction

High-quality scientific experiments require precise laboratory equipment. Experimental devices designed for studies of mechanical interactions typically contain force sensors and positioning motors. An example of a high-precision experimental setup that operates with mechanical contact is an atomic force microscope (AFM), which has lateral resolution up to a few nanometers and a normal resolution up to 0.1 nm. AFM technology also uses force sensors, which allow for measurement of van der Waals forces in the range of 10^−10^ to 10^−12^ N [1,2,3]. Experimental facilities that deal with nanotechnology often use a variety of high-sensitivity electrical, mechanical, optical and other sensors to measure physical quantities of low magnitude [4,5].

High-accuracy force measurements are very important in macroscopic experiments with mechanical contacts, especially in cases in which the smallest changes in macro parameters need to be detected. For investigation of mechanical contacts of adhesive surfaces, high-precision force and positioning sensors are used as main components [6,7,8]. A number of tribometers and force sensors with a variety of measuring features have been designed recently for this purpose [9,10,11,12,13,14,15,16,17]. A central part of these experimental setups are force sensors manufactured by various electronics companies (see, for example, [18]). Many of them are compatible with personal computers (PCs) and other laboratory equipment through standard interfaces and can be easily adopted to measure desirable quantities. However, to create a fully operational device that performs complete cycles of specific experiments, sensors, motors and other parts must be assembled into one apparatus. There are various potential sources of errors in complete systems that have to be compensated for, e.g., signal drift, temperature fluctuations and hysteresis in the force sensor [14]. 

In this paper, we provide an overview of laboratory setups equipped with KD24S and K3D40 force sensors manufactured by ME systems, which were designed to perform experiments on the indentation of a rigid indenter into a soft substrate [19,20,21]. The designed facility operates as a high-precision tribometer. It performs indentation and retraction in the normal direction, as well as displacement in the tangential direction, with precise positioning of the sample in all three dimensions, in addition to measuring loading forces, enabling observation of dynamical processes occurring in the contact area in real time with photo and video recording. 

During a series of experiments, the designed device was upgraded several times according to progress in research plans. Thus, the earliest version of the apparatus was able to perform experiments only with normal contact, whereas the current version operates with three-dimensional positioning. Based on our experience with the design and use of the developed equipment, herein, we outline its pros and cons and discuss the well-known peculiarities of tribometers, such as backlash in motor and signal drift in the force sensor. We also suggest some practical solutions that helped us to overcome certain issues in the performed experiments. 

Our work will be interesting to scientists and engineers who work in the field of contact mechanics and deal with similar instruments. 

## 2. Materials and Methods

In our experiments, we studied physical phenomena within mechanical contacts that involve highly adhesive surfaces. Adhesion often plays a crucial role in the natural environment and numerous technologies. As an illustration of adhesive processes in a biological system, we can refer to the ability of gecko lizards to climb vertical surfaces and move on ceilings [22,23,24]. Adhesion is also important for microorganisms, viruses and living cells [25,26,27]. The study of specific adhesion processes is an important part of research on novel SARS-CoV-2 (COVID-19) coronavirus infection [28]. Technologies based on adhesive phenomena have found numerous applications in industry, where various surfaces come into contact, such as coatings, welding, granulation and other applications [29,30].

In order to study contact between a rigid body and an elastomer under laboratory conditions and obtain insights into the physical processes in the contact area, an appropriate laboratory setup was designed and fabricated. The concept of the designed laboratory device was inspired by an instrument developed in [19]. Figure 1 shows the progress in designing and assembling the facility, from the idea of a device pictured as a sketch (Figure 1a) to an exactly matching 3D model of all its parts (Figure 1b) and, eventually, a fully operable piece of equipment (Figure 1c). All functional parts of the device that are denoted in the figure are the same for all three panels, namely (1) and (2) high-precision M-403.2DG motorized linear stages (manufactured by PI), which are handled by PI C-863 one-axis servo controllers; (3) an ME K3D40 three-axis force sensor; (4) an indenter that is mounted on the force sensor; (5) the sample being indented; (6) a tilt mechanism; and (7) a digital camera. A GSV-1A4 SubD37/2 four-channel analog amplifier was used to amplify the electrical signal from the force sensor. The amplifier is coupled with a computer via an NI USB-6211 16-bit analog-to-digital converter (ADC). A photo of the contact region with a clear view of the indenter (4), force sensor (3) and sample (5) is shown in Figure 2 (note the different configuration of the experiment and indenter compared to Figure 1c). In our experiments on adhesive contacts we often use optically transparent rubber sheets produced by Japanese company TARNAC (see [31]) as a substrate, because this rubber is optically transparent and has high adhesive properties. As shown in the Figure 1c, the device has a cylindrical steel indenter (wheel). This particular device was used to perform experiments investigating adhesion in rolling and sliding friction in [32]. Moreover, equipment with different indenters has been used to perform a series of experiments [20,21,32,33,34,35,36,37,38,39].

The developed device is shown in Figure 1c. The main disadvantage of this device is the backlash in the M-403.2DG motorized linear stages. These drives are capable of high-precision positioning (up to 1 µm) of the indenter in unidirectional repeatability mode; however, after changing the direction of motion, their backlash can reach 10 µm. In our experiments, we observed backlash of about 6 µm for both drivers (1) and (2) (see Figure 1). Technically, this means that after changing the direction of motion, the device shows that the indenter is still moving (the screw attached to the motion platform is rotating), even though for the first six micrometers, the indenter remains physically motionless. This effect must be considered during data analysis; otherwise, it will lead to inadequate experimental results for both indentation and pull-off. To illustrate the impact of a backlash on experimental data, let us consider the following example.

Figure 3 shows the results of an experiment in which a steel sphere with a radius of *R* = 33 mm was indented into a layer of TARNAC CRG N3005 rubber with a thickness of *h* = 5 mm. After the indentation depth reached its maximum value of *d*_max_ = 0.2 mm, the indenter was moved in the opposite direction until it lost contact with the substrate. The curve plotted in red represents the indentation phase. It starts from zero and is located above the pull-off curve, which is indicated in blue. Owing to the strong adhesive interaction between the steel indenter and rubber substrate, negative forces of *F_N_* < 0 N are present in the system during detachment. This means that additional force must be applied to the indenter to detach it from the sample compared to contact force at the same position of an indenter during the indentation phase. The presence of backlash in the M-403.2DG drive leads to a situation in which the indenter physically remains in a stationary position, whereas the equipment shows its motion for six micrometers after the change in direction. Moreover, the magnitude of normal force (*F_N_*) remains constant, as there is no actual movement in the contact. Therefore, the pull-off curve becomes shifted to the left along the abscissa axis and crosses the loading curve related to indentation at point *A* (Figure 3a).

To obtain an accurate diagram of a contact load, backlash correction must be introduced in data analysis. In our case, such a correction is a simple addition of the compensational term of 6 µm (backlash magnitude) to the pull-off curve, which shifts it along the *x* axis in the positive direction, as shown in Figure 3b. In general, the need for data correction can be determined according to the goal of the experiments. If the goal of an experiment is to determine the adhesive strength of a contact (minimal external force needed to completely destroy the contact), as in [20], then adjustment is unnecessary, as the shift along the abscissa axis does not change the magnitudes of the acting force. On the other hand, if the dissipation of the mechanical energy in the full cycle of loading during indentation and pull-off (see also [20]) has to be determined, then correction plays a crucial role, as the dissipation is defined by the area of a region bounded by both indentation and pull-off curves (*F_N_*(*d*)). However, such data corrections are not always possible. For instance, the presence of backlash makes the interpretation of the results of experiments even more difficult when the indenter performs a cycling motion (see [34]).

Figure 3 illustrates a simple adjustment of the data by shifting to the right the pull-off curve, and it is completely sufficient to obtaining an adequate result from the experiment. However, assuming that our experiment requires changing the direction of motion more than one time during the pull-off phase prior to complete disruption of the contact, we would observe a backlash that starts at the turning point and lasts for 6 µm as the indentation depth (*d*) increases according to the equipment measurements. Thus, the obtained curve must be shifted left instead of right. As a result, during cycling motion, dependence related to a change in direction from indentation to pull-off must be shifted to the right, whereas dependence that corresponds to the change from pull-off to indentation must be shifted to the left. As a result of such data manipulations, the total load curve (*F_N_*(*d*)) becomes non-closed. Technically, this occurs because, owing to the backlash, the indenter is not physically present in certain segments of its expected trajectory. Therefore, analysis of the *F_N_*(*d*) curves related to the cycling load requires extrapolation of the obtained dependencies, which may significantly affect results. Studying oscillations with small amplitudes becomes virtually impossible, as within the backlash magnitudes, the indenter does not move at all.

It is worth mentioning that backlash magnitude depends on the temperature and load applied to the driver, so the automatic correction of this particular feature is a very complicated challenge, and every new load scenario requires additional analysis and corrections. One possible solution to the described problem is a direct measurement of the indenter coordinate by a laser vibrometer; however, this dramatically increases the cost of the equipment, as well as its size. Another solution is to use different types of drives that have improved positioning mechanisms without backlash. Although such drives are more expensive than the M-403.2DG described above, they are cheaper than using a laser vibrometer or similar high-quality equipment. In the upgraded version of the developed equipment, we adopted the second solution (see description below).

Experience reveals another peculiarity of the equipment shown in Figure 1 related to the Conrad USB digital camera. This camera has a resolution of 1600 × 1200 pixels, which is enough for precise monitoring of the contact area. However, a disadvantage of this camera as a part of the designed tribometer is the brightness correction, which is performed automatically if the intensity of external illumination changes. In our experiments, the indenter moved in the normal and tangential directions, as well as in both directions simultaneously; as a result, the distribution of luminous flux was affected. Under stationary conditions, illumination of the sample is homogeneous, as it is provided by a static comprehensive lighting system (see Figure 1c and Figure 2). Light redistribution causes the camera to switch into a new mode of photographing, and as a result, pictures taken of the contact area have different contrast. This situation does not change the general view of the contact zone; however, it makes automatic processing of the photos and determination of the size of the contact area much more complicated [20,33,34]. The software developed to analyze the digital photos of the indentation zone and to calculate the contact area uses an algorithm that calculates the size of a contact by analyzing the difference in the intensity of pixel color, which determines whether a given pixel is included in the contact area or not (for a detailed description of the software performance, see [20]). If the intensity of pixels constantly changes due to the different settings of the camera mode, such analysis becomes extremely difficult.

Assembling the equipment shown in Figure 1 was a significant advancement from the initial facility [19], as the options for tangential motion of the indenter and more precise positioning of the indented sample, as well as the possibility of considerably increasing lateral sizes of the substrate, were added to the device. However, as mentioned above, the device shown in Figure 1 has its downsides as well. With the goal of rectifying these disadvantages and increasing the quality of the obtained results, the equipment was upgraded. An exact 3D model of the upgraded equipment and a photo of the assembled device are shown in Figure 4a,b, respectively. As shown in the figure, the device has several differences from the previous version.

The main novelties of the proposed equipment are:(a)More precise L-511.24AD00 drives with bidirectional repeatability up to ±0.2 µm. The main attribute of these drives is the absence of backlash after changing the direction of motion. This feature is achieved using a ball-screw transmission and an incremental linear encoder inserted for feedback. The encoder performs direct measurements of the moving platform coordinate during motion and corrects it. Thus, these drives do not create artefacts after changing the direction of motion and have high-precision positioning that is more than enough for our experiments. However, the presence of feedback makes the positioning system vulnerable to equipment vibrations. These vibrations could prevent the correct positioning of the motion platform, as the drive repeatably searches for correct coordinates and, as a result, works as a generator of mechanical oscillations, making measurements impossible. Therefore, an oscillation damper for the whole device must be introduced to the facility. To dampen oscillations of the equipment, we used rubber (TARNAC CRG N0505 [31]) sheets placed between the stands of the device and the laboratory table with pneumatic support (see Figure 4b), serving as a foundation on which the device was assembled. That particular rubber is optically transparent and has highly adhesive properties, so it was typically used as a substrate for indentation in our experiments, as it allows for direct observation of the processes that occur in the contact area through the layer of rubber via a camera. However, this material was initially designed as an oscillation damper [31], so in this case, it was used as an intended. It is also worth mentioning that maximal velocity of the table movement in L-511.24AD00 drives is 90 mm/s, which is significantly higher than the 2.5 mm/s in M-403.2DG.(b)A high-quality XimeaMQ022CG-CM color camera with a FUJINON HF16SA-1, 2/3” lens. This camera is USB 3.1-compatible and has a resolution of 2048 × 1088 pixels at 170 frames per second, making observation of fast processes in the contact area possible. In our future experiments, we plan to use it to study the propagation of elastic waves in the contact area during its reconfiguration. The presence of these waves was detected in experiments with the bare eye; therefore, more detailed study with special equipment is needed. Another advantage of this camera as a part of the designed tribometer relative to previously used cameras is its manual brightness setup instead of an automatic mode. The brightness is regulated by a built-in mechanical diaphragm; as a result, all taken photos have the same brightness level, which makes it easier to automatically calculate the contact area using image analysis software.(c)A large STANDA 8MR190-90-4247-MEn1 motorized rotation stage (position (8) in Figure 4a,b) operated by a single-channel 8SMC5-USB-B8-1 controller was added to the facility, which allowed the horizontal orientation of the substrate to be changed. The maximal angular speed of the stage is 36°/s (6 full circles in a minute). This stage has a clear aperture of 89 mm, which is almost twice as large as the 45 mm of the previous version of the equipment, and allows for observation of contact processes in a wider range of tangential displacement of the indenter. According to its specification, the stage is characterized by a backlash of 40 arcsec and bidirectional repeatability of 72 arcsec, which are related to the displacement at the edge of an aperture 8.67 µm and 15.54 µm, respectively. These features must be taken into account during experiments in which the direction of circular motion changes. However, this was not an issue in our experiments. The rotation stage was added to the facility with the intention of studying a stationary sliding mode without changing the rotation direction.(d)A completely new tilt mechanism (position (6) in Figure 4a,b). The biggest difference between the upgraded version and the previous mechanism shown in Figure 1 is a much wider aperture (equal to the 8MR190-90-4247-MEn1 stage aperture), which was enlarged to observe larger portion of the contact area. We could not find the required in the available markets, so it was completely designed and manufactured in-house at the TU Berlin workshop from aluminum plates. The 3D blueprint and a photo of the fabricated mechanism are shown in Figure 4c.

The device shown in Figure 4 was designed in a such way that drives (1) and (2), together with the rotation stage (8), are capable of positioning the indenter over the substrate in three dimensions (as shown in Appendix A). Although described equipment was designed for experiments on adhesive contact with complex indentation trajectories, it can also be used for other purposes depending on what is installed as an “indenter”. For instance, if a laser emitter is mounted instead of an indenter, the device will work as a basic laser engraver. Similarly, a solid ink printhead transforms the device into a simple 3D printer, and with any writing tool, the equipment operates as a plotter. All mentioned (and similar) modifications can be introduced to the facility without any significant changes to the main part of the equipment. To demonstrate the possibility of modification, we created a simple analog of a plotter using an ordinary pen as a writing tool. An example of its performance is presented in Appendix A. 

It is worth noting that a further important aspect relates to measurements. Experiments described in the original work [19] were performed with an ME KD24S force sensor operating within the range of ±2 N. The signal from this sensor was amplified by a single-channel strain gauge amplifier/DM1 DMS amplifier (manufacturer LEG–Industrie-Elektronik GmbH). Furthermore, the upgraded version of the facility (Figure 1 and Figure 4) was equipped with an ME K3D40 three-axillar force sensor (version with a force range of ±10 N) and a GSV-1A4 SubD37/2 4-channel amplifier. In all versions of the device, the output signal from the amplifier was transmitted to a personal computer through the NI USB-6211 16-bit ADC. Many force sensors (including that used in our equipment and described above) have signal drift, due to which even unloaded sensors display increasing (or decreasing) force over time. Figure 5 shows the time dependencies of force measured by unloaded (i.e., the magnitude of actual force acting on the sensor during the whole measurement equals zero) sensors.

Dependencies in panel (a) were measured by the ME KD24S sensor. As shown in the figure, apart from three bottom lines, the signal drift is almost linear, with different slopes. Linear drift can be easily subtracted from obtained data; however, the presence of non-linear progressive changes in the signal make data analysis more complicated. Figure 5b shows similar data but measured by another ME K3D40 sensor. Here, all three components of a force were measured in one experiment. As the figure shows, the signal drift is also present in the measurements; however it is significantly lower than in most data measured by the ME KD24S sensor (see Figure 5a).

Thus, sensors with small or no drift are always preferable as a part of laboratory equipment (for instance, ME K3D40 instead ME KD24S). Furthermore, the impact of signal drift on a measured force magnitude can be reduced by various methods. For instance, it can be subtracted from the measured data if the shape of the signal drift is known. Another way to reduce drift influence is to perform the indentation cycle as fast as possible or measure the data within the range of forces that significantly exceed the drift magnitude so it will not affect the results. However, these methods often contradict the required conditions of the experiments. It is worth noting that dependencies shown in both panels of Figure 5 were obtained using a GSV-1A4 amplifier; however, in experiments with a DMS DM1 amplifier, similar problems with drift signal were observed. Additionally, in our experiments, two different ADCs (NI USB-6211 and NI USB-6000) were used for the PC interface, and signal drift was observed with both of them.

Let us consider an illustrative example of how to handle signal drift from the sensor and to correct the results. Among the two available sensors, ME KD24S has the larger signal drift; thus, we will demonstrate the data processing procedure with this sensor. Figure 6 shows the measured dependencies of the normal force (*F_N_*) on the indentation depth (*d*).

The presented dependencies were measured in an experiment on indentation of a steel sphere with a radius of *R* = 30 mm into a highly adhesive TARNAC CRG N0505 rubber substrate. In the experiments, the indenter reached an indentation depth of *d*_max_ = 0.1 mm, then moved in the opposite direction up to *d*_min_ = −0.2 mm. Complete detachment of the indenter was observed in all experiments before it was pulled off to its highest position above the substrate (*d*_min_). It is worth noting that none of the dependencies shown in Figure 6 exhibit hysteresis after a change in the direction of motion (compared to the data presented in Figure 3) because of the L-511.24AD00 backlash-free drives used in the upgraded equipment. The points of complete detachment during the pull-off phase can be detected for all dependencies shown in Figure 6a. On the bottom curve, it is denoted as point *A*. Although there was no contact at point *A* and the indenter was completely detached from the substrate, a non-zero normal force was detected by a force sensor due to its signal drift. The magnitude of this force and the related indentation depth are denoted as *F_A_* and *d_A_*, respectively. If during the experiment, drift is considered to be linear (as in top the 7 dependencies shown in Figure 5), then the corrected time dependence of the normal force (*F_N_*(*t*)) can be easily defined as:(1)$${\tilde{F}}_{N}\left(t\right)=F_{N}\left(t\right)-\frac{F_{A}}{t_{A}}t,$$
where *t_A_* is the time interval from the beginning of the experiment until complete detachment (point *A*). In the experiments shown in Figure 6, the velocity of the indenter was chosen in such a way that the full cycle of indentation for every curve lasted for 30 min, similarly to the signals shown in Figure 5. Thus, the path of the indenter during the measurements was *s* = 2*d*_max_ + |*d*_min_| = 0.4 mm, whereas its velocity was *v* = *s*/*t* = 0.4/1800 ≈ 2.22∙10^−4^ mm/s or 0.222 µm/s. At this velocity magnitude, the contact can be considered quasi-static [35]. Dependencies corrected according to procedure (1) are shown in Figure 6b. As shown in the figure, all curves except the bottom line almost overlap in the indentation phase. Therefore, we can conclude that the bottom curve cannot be corrected through expression (1), as the signal drift during its measurements was non-linear (see also the three bottom lines in Figure 5a). The difference between three upper curves in Figure 6b during the pull-off phase can be explained by adhesion phenomena. Thus, adhesion slightly affects contact propagation during the indentation phase, whereas during detachment, it plays a crucial role [19,20,33,34,35]. During the indentation phase, dependence (*F*(*d*)) is determined only by elastic parameters (elastic modulus (*E*) and Poisson’s ratio (*ν*)) of the rubber layer. These parameters determine the rigidity of a contact and are constant over time. Moreover, the specific work of adhesion significantly affects the pull-off phase. The specific work of adhesion, as a function of surface energies of contacting bodies, decreases for each new indentation cycle because of dirt and oxidation on the surfaces [21]. Therefore, the differences between the three upper *F_N_*(*d*) curves shown in Figure 6 in the pull-off phase are the result of natural causes that are not related to signal drift.

Figure 6 shows the results of an experiment on indentation of a smooth spherical indenter with monotonic *F_N_*(*d*) dependencies. However, if the indenter has noticeable roughness or an irregular geometric shape, the dependence of a normal force on indentation depth may be non-monotonic during the pull-off phase [20,21]. Therefore, it is important to understand whether such drastic changes in force are caused by physical processes in the contact area or whether these changes are measurement artefacts (an example of such an artefact is the third curve from the bottom in Figure 5a). For this purpose, it is necessary to analyze additional characteristics such as the configuration of the contact area. Any sharp changes in force (*F_N_*) are, naturally, caused by reconfiguration of the contact area; thus, such reconfiguration must be observable. In our experiments, used a transparent substrate, which allows for direct observation of the contact area. Appendix A shows an example of such an experiment (corresponding measurements are shown in Figure 6). In the video, both the original and corrected dependencies (*F_N_*(*d*)) are shown, together with images of the contact area. It is worth mentioning that this example shows an experiment with a mirror-polished indenter; thus, the camera, lights and other equipment parts are reflected on its surface. This makes automatic calculation of the size of the contact area based on numerical analysis of digital photos impossible. Therefore, if the information on the size of a contact area is critically important (for example, for calculation of shear stresses), an indenter with low-amplitude roughness is preferred for experiments. In this case, the contact area can be identified more easily due to diffuse light scattering, and its size can be calculated automatically by software through analysis of the taken photos (for example, see [21] and related Appendix A).

Figure 7 shows the dependencies measured in the experiment under the same conditions as the data presented in Figure 6, only with a K3D40 force sensor. As the K3D40 force sensor has significantly smaller signal drift compared to the KD24S sensor (see also Figure 5), the dependencies (*F_N_*(*d*)) that are shown in Figure 7a already depict the contact processes precisely enough. Removing the drift does not significantly affect the quality of the obtained results (see Figure 7b). It is worth noting that installed K3D40 force sensor operates within a force range of ±10 N; thus it is not as precise as the KD24S sensor (version with a range of ±2 N). Therefore the *F*(*d*) dependencies measured by a K3D40 sensor exhibit a significant amount of noise. Nevertheless, in our experiments, the K3D40 force sensor was the preferred choice over the KD24S, as it does not require additional analysis of the obtained data or compensation for signal drift.

## 3. Data Analysis and Interpretation

The Figure 3, Figure 6 and Figure 7 presented above contain the results of experiments involving normal indentation of steel spherical indenters into a soft elastomer. A general feature of all performed experiments is the difference between *F_N_*(*d*) dependencies related to the indentation phase and pull-off. Such behavior is typical for adhesive systems and assumed to be caused by several factors: roughness of the surfaces, viscoelasticity and the humidity effect, among others [40]. The exact cause of hysteresis after changing the direction of indenter movement has not been clearly established yet. JKR theory [41] is frequently used to describe the adhesive contact during the indentation of rigid spheres into soft elastomers. JKR determines the relations between the normal force (*F_N_*), indentation depth (*d*) and contact radius (*a*) as [41]:(2)$$d\left(a\right)=\frac{a^{2}}{R}-\sqrt{\frac{2\pi a\Delta \gamma }{E^{*}}},$$
(3)$$F_{N}\left(a\right)=\frac{4E^{*}a^{3}}{3R}-\sqrt{8\pi a^{3}E^{*}\Delta \gamma },$$
where Δ*γ* is a specific work of adhesion, and *E*^*^ = *E*/(1–*ν*^2^) is a reduced elastic modulus; parameters *E* and *ν* is the elastic modulus and Poisson’s ratio of the elastomer. According to Equation (2), after the first contact at *d* = 0 mm, the contact area immediately propagates in size up to equilibrium radius
(4)$$a\left(d=0\right)={\left(\frac{2\pi R^{2}\Delta \gamma }{E^{*}}\right)}^{1/3},$$
where the corresponding adhesive force, according to Equation (3), is negative:(5)$$F_{N}\left(d=0\right)=-\frac{4}{3}\pi R\Delta \gamma .$$

The experiments described above consider the indentation of a steel indenter into a soft substrate, so the indenter can be considered as absolutely rigid, i.e., no strains occur in the indenter. However in the case in which the indenter undergoes deformation, relations (2) and (3) can still be used to describe the contact mechanics if the reduced elastic modulus is defined as 1/*E*^*^ = (1–$\nu_{1}^{2}$)/*E*_1_ + (1–$\nu_{2}^{2}$)/*E*_2_, where indices 1 and 2 denote the materials of the indenter and half-space, respectively. It is worth noting that expressions (2) and (3) are valid within the half-space approximation when the contact radius (*a*) is significantly smaller than the thickness of the elastomer (*h*); otherwise, the elastomer thickness (*h*) must be taken into account. Several methods are available for this procedure, including, for instance, analytical [42] or numerical [43] modeling.

Figure 8 shows the *F_N_*(*d*) dependencies obtained in our experiments and discussed above. Each panel of the figure shows one cycle of indentation; Figure 8a relates to Figure 3b, Figure 8b relates to Figure 6b, and Figure 8c relates to Figure 7b. Besides the experimental dependencies (solid lines), Figure 8 also shows the results of a numerical simulation (dashed lines) performed according to the boundary elements method (BEM), where the thickness of the elastomer (*h*) is also taken into account [43]. In the BEM model, main parameters such as indenter radius (*R*), elastomer thickness (*h*) and indentation depth (*d*) are exactly the same as in the experiment.

In the simulation, we assumed that the specific work of adhesion (Δ*γ*_1_) during the indentation phase was significantly smaller than Δ*γ*_2_ during the pull-off. The choice of different magnitudes of Δ*γ* for indentation and pull-off is a rather phenomenological approach; nevertheless it enables description of the behavior of the system that is observed in experiments, although the exact reasons of this behavior remain unknown for now. As follows from Figure 8, the results of simulations and experiments almost overlap in the indentation phase, whereas during the pull-off, there are clearly noticeable differences. The main reasons for these differences are the simulation conditions; the indenter has an ideal spherical shape, and the elastomer is absolutely flat. Moreover, the adhesion-specific work (Δ*γ*) is evenly distributed over the surface (the same value at every point), and factors such as surface roughness, viscoelastic effects and others are not taken into account.

In numerical simulations for CRG N3005 material (Figure 8a), the following values of elastic parameters were used: *E* = 0.31 MPa and *ν* = 0.47, Δ*γ*_1_ = 0.027 J/m^2^ during the indentation phase and Δ*γ*_2_ = 0.19 J/m^2^ during pull-off. For CRG N0505 material (Figure 8b,c) the corresponding values were: *E* = 0.031 MPa, *ν* = 0.47, Δ*γ*_1_ = 0.027 J/m^2^ and Δ*γ*_2_ = 0.24 J/m^2^. In both cases, the magnitude of Δ*γ*_1_ related to the indentation phase is significantly smaller than Δ*γ*_2_ related to pull-off. However, even a low value of Δ*γ*_1_ ensures the propagation of the contact due to adhesion. After the moment of the first contact, the contact area expands (see Appendix A), and the normal force is negative (*F_N_*(*d* = 0) < 0 N). Negative force at zero indentation depth (*d*) during the indentation phase is clearly visible in Figure 8b,c. Dependencies in Figure 8a are shown in a wider range of a normal force so that the mentioned behavior is not visible; however it is clearly visible in the inset. Note that the adhesive strength of the contact (the maximum absolute value of the adhesive force during pull-off at *d* < 0 mm) is approximately the same for all cases and equals *F_A_* ≈ 30 mN. In the case of half-space when the JKR theory (2), (3) is valid, the maximum of the adhesive force (corresponding to the minimum of the *F_N_*(*d*) dependence during pull-off) is defined as:(6)$$\left|F_{\min}\right|=\frac{3}{2}\pi R\Delta \gamma .$$

If a rigid sphere is indented into a relatively thin layer of elastomer (the contact radius is significantly larger than the elastomer thickness), the adhesive strength increases up to a magnitude of |*F*_min_| = 2*πR*Δ*γ* in the case in which the elastic film is frictionless on the interface between the film and the substrate. When the elastic film is perfectly bonded on the interface between the film and the substrate, the adhesive strength became even larger, i.e., |*F*_min_| = 3*πR*Δ*γ* [44,45]. Therefore, in the general case, adhesive force can be defined as |*F*_min_| = *απR*Δ*γ*, where parameter *α* depends on the condition of the experiment. For the dependencies shown in Figure 8, Δ*γ* ≈ 0.2 J/m^2^, *R* ≈ 30 mm and |*F*_min_| ≈ 30 mN, so α ≈ 1.6, which is very close to the JKR estimation (6).

## 4. Investigation of the Tangential Contact

In the proposed work, we describe the design and performance of the facility, the main purpose of which is to study tangential contacts. In the previous sections, we considered typical problems of the equipment (backlash of the drives, drift of the signal from the force sensors and the peculiarity of the lighting system) and suggested possible solutions. However, all the examples of device performance presented above were related to normal indentation. Hence, to complete our description, in this section, we consider an experiment involving tangential motion. In the mentioned experiment, an indenter with a radius of *R* = 40 mm was indented into a layer of rubber to a depth of *d* = 0.4 mm, followed by a shift in the tangential direction for *x* = 5 mm before returning to the previous position of *x* = 0 mm. Then, the indenter was lifted up in the normal direction to complete detachment.

The results of the experiment are shown in Figure 9. The figure shows time dependencies of all three components of the contact force, along with images of the contact area taken at certain moments of time.

In general, the presented dependencies are similar to the data described in our previous work [37]. Thus, after some time, the system exhibits a sliding mode with constant friction force (*F_x_*). As the indenter moves only along the *x* direction, the lateral component *(F_y_*) is expected equal zero in an ideal case; however, we observed non-zero values of this component. This situation arises for several reasons, such as non-precise positioning of the indenter, inhomogeneities in the contact area, etc. In real experiments, *F_y_* should be small enough compared to *F_x_*. As shown by the presented dependencies, this condition is fulfilled in our experiments. The main difference between the experiment shown in Figure 9 from that described in [37] is the chemical inhomogeneities on the surface of the indenter.

Inhomogeneous distribution of the surface energy was created by a short (approximately 1 min) treatment of the surface of the steel indenter with a water solution of FeCl_3_; then, the indenter was washed with water, then washed with alcohol and dried. Treated areas of the indenter can be seen as dark spots in the photos shown in Figure 9. These spots are characterized by a stronger adhesive strength compared to the untreated area [21]. As it follows from the Figure 9, inhomogeneous distribution of the surface energy leads to more complex contact behavior; elastic waves and pores appear in the contact area, and contact is no longer simply connected. Figure 9 shows only part of images of the contact area, which are not enough to track the dynamics of the tangential shift. A complete illustration of this process is presented in the Appendix A. The description and analysis of this processes is a non-trivial task, which is not the aim of this work and will be provided elsewhere.

It is worth noting that behavior similar to that described above was observed in experiments more than 50 years ago; waves that propagate within the contact area during tangential shift are known as Schallamach waves [46]. Although several theoretical and experimental studies of Schallamach waves have been conducted (see for example [47,48,49]), no complete explanation of the process exists yet. This situation is caused by the fact that well-known and popular theories such as JKR [41] are built for quasistatic contact and cannot be applied for description of fast dynamical processes. Moreover, as experiments show, the most complex behavior during wave propagation is observed at high values of the adhesion-specific work, as in this case, the rubber undergoes severe non-linear deformation prior to propagation of the first wave. Additional factors, such as liquid or particles in the contact area, destruction of the material, chemical inhomogeneities (see Figure 9 and Appendix A), etc., also affect wave propagation. In our future works, we plan to study the behavior shown in Figure 9 more precisely. 

## 5. Summary and Discussion

Various modifications of the described device were used in several experiments, the results of which have already been described in the literature [19,20,21,32,33,34,35,36,37,38,39]. In particular, in [19], experiments on contacts of rigid punches that have a flat but oddly shaped face with a soft adhesive layer were performed; in [20], the influence of the roughness of contacting surfaces on adhesive strength was studied; surfaces with artificially created chemical inhomogeneity were studied in [21]. Additionally, in [21], the presence of a third body in the contact area was considered. This particular case simulates wear debris in contacting parts of various mechanisms. Peculiarities of the sliding and rolling friction modes were studied in [32]; in [33], the dependence of adhesive strength on the duration of the contact was investigated; dissipation of the mechanical energy within oscillating adhesive contact was studied in [34]. Therein, it was shown that dissipation is not observed if the oscillation amplitude does not exceed a certain critical magnitude. In [35], an experimental study of the secondary adhesive hysteresis that occurs after changing the direction of motion of the indenter was conducted; contact of ellipsoidal flat forms with an elastomer substrate was studied in [36]. In particular, the influence of eccentricity of the ellipsoidal form on contact processes was investigated. How adhesion affects sliding friction (tangential contact) was studied in [37]. The transportation of particles driven by surfaces with periodic relief in tangential contact was studied in [38]. All experiments mentioned in this paragraph involved adhesive contacts.

It is worth mentioning that the sensitivity of the designed equipment is limited by the abilities of the installed force sensor, which can be increased by using more capable sensors. Furthermore, although functional modification of the device for certain types of experiment is possible, more advanced studies or experiments under unique conditions require a special type of tribometer. As notable examples of such tribometers, we can refer to a millitribometer designed to study small friction units and small loads [50], a six-axis force/torque sensor for slow friction measurements under ultra-high vacuum and other conditions [51] and a setup for the study of friction-induced vibrations [52] and haptics [53].

Application of the designed facility is not limited to studying adhesion. Thus, experiments performed in [39] considered capillary forces and their effect on the strength and shape of a contact. For this purpose, a drop of liquid was placed between contacting surfaces (a steel sphere and a glass sheet).

In the articles cited above, the designed equipment was used to study both normal and tangential contacts. However, as we described it in the previous section, a motorized rotation stage was recently on the facility. This particular upgrade adds additional degrees of freedom by making a rotation of the indented sample possible. Therefore, the upgraded setup will be used in our future experiments on the transition between friction and adhesion modes. Our previous experiments on tangential motion revealed the presence of a sliding mode with constant tangential mechanical stresses in the contact area. In this mode, the friction force depends on the size of the contact area, whereas the friction coefficient depends on the indentation depth; in this case, the friction coefficient cannot be considered a parameter characterizing friction. This is a well-known and typical behavior of friction contact of surfaces with high adhesion interaction [37,54]. However, in [54], it was experimentally shown that in such contacts, the friction mode (which can be characterized by a friction coefficient) is observable under large loads. Thus, in upcoming experiments, we plan to study the transition between two the mentioned modes by gradually increasing the indentation depth (contact load). Such experimental an condition can be achieved with the installed rotation stage. The rotation stage can provide the possibility of performing formally infinite motion, as it is not limited by the sliding range of a linear stage. 

The designed facility will also be used in experiments with more complex load scenarios, in which contact modes with non-trivial behavior may occur. For instance, in a theoretical study [55], it was shown that the friction coefficient in a dry contact might depend on load history if rotation is present in the system. We plan to verify these particular theoretical results experimentally.

## 6. Conclusions

Herein, we proposed an intuitive strategy for designing and assembling a laboratory tribometer that uses a force sensor as the main measuring tool. The proposed approach based on previous experience concerning design of laboratory equipment and conducting tribology and contact mechanics experiments. We also discussed particular problems and suggested possible solutions that emerged during the use of the equipment. In particular, the signal drift of force sensors and the backlash of the indenter drives was discussed with respect to a number of measurements.

## Figures and Tables

**Figure 1 sensors-23-00456-f001:**
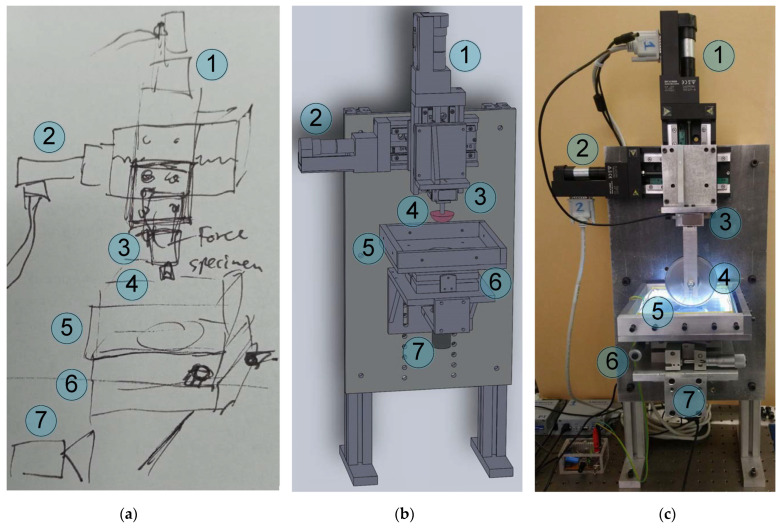
The process of development of the laboratory equipment: design starts with the concept, shown as a sketch panel (**a**), which later is transformed into a detailed blueprint panel (**b**) and then assembled as a laboratory tribometer panel (**c**). The labeling of the equipment is the same in all panels: (1) and (2) M-403.2DG high-precision motorized linear stages (manufactured by PI), (3) three-axis force sensor ME K3D40 with (4) a mounted indenter, (5) the sample being indented, (6) a tilt mechanism and (7) a digital camera. (**b**,**c**) Device with different indenters.

**Figure 2 sensors-23-00456-f002:**
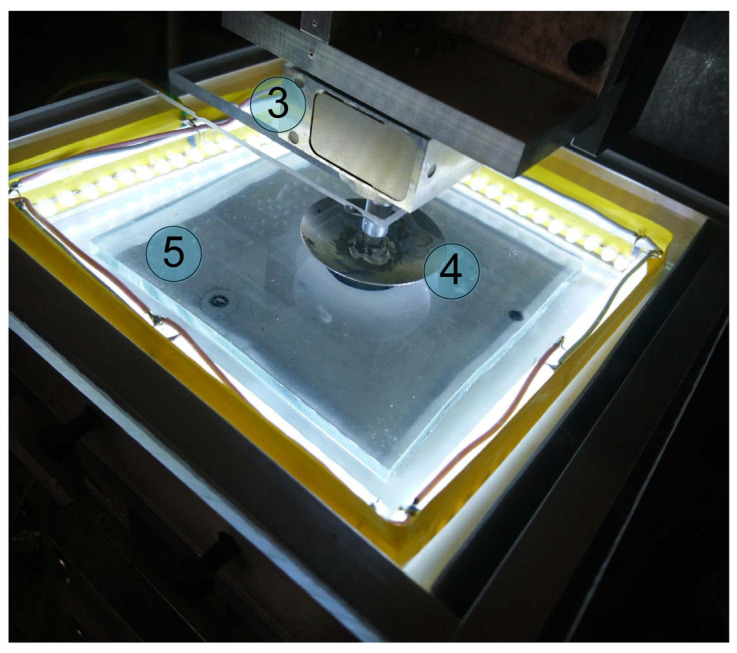
Detailed photo of the contact region: indenter (4), sensor (3) and elastomer (5). The all-sides lighting system consisting of 80 LEDs is also clearly visible.

**Figure 3 sensors-23-00456-f003:**
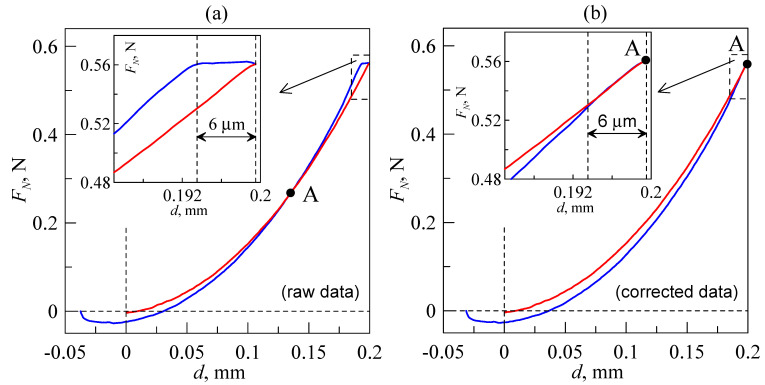
Dependencies of the normal force (*F_N_*) on indentation depth (*d*) measured during the indentation of the spherical indenter with a radius of *R* = 33 mm into a rubber (*TARNAC CRG N3005*) substrate with a thickness of *h* = 5 mm [33]. Experiments were performed with the equipment shown in Figure 1. (**a**) Raw measured data; (**b**) dependencies with compensation for a backlash of the drive.

**Figure 4 sensors-23-00456-f004:**
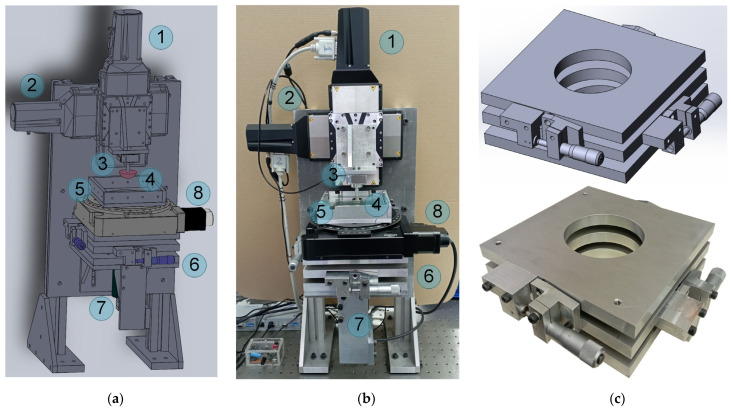
Upgraded version of the device shown in Figure 1. (**a**) Exact 3D model of the facility with an individual blueprint for every part. (**b**) Photo of the assembled and fully operational equipment. (**c**) A 3D model (top) and photo (bottom) of the tilt mechanism fabricated from aluminum plates, which is one of the most significant improvements added to the device. Position (8) denotes the 8MR190-90-4247-MEn1 motorized rotation stage in (**a**,**b**); all other labels are the same as in Figure 1.

**Figure 5 sensors-23-00456-f005:**
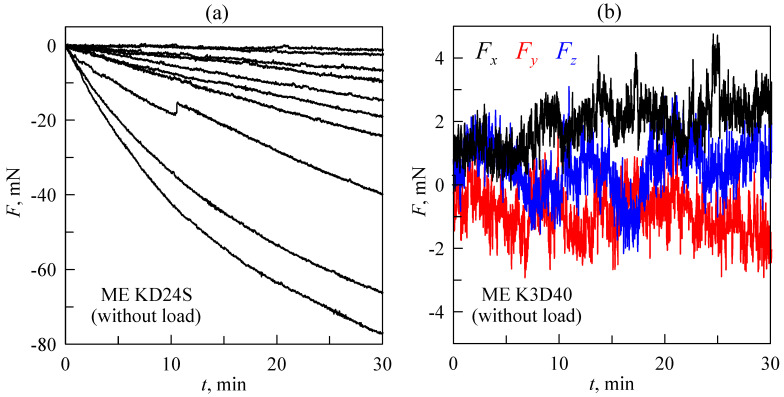
Time dependencies of “force” measured by an unloaded sensor (signal drift). (**a**) Ten dependencies corresponding to different measurements obtained by the ME KD24S sensor. (**b**) Time dependencies of three components of a force measured by the ME K3D40 three-axis sensor in one experiment.

**Figure 6 sensors-23-00456-f006:**
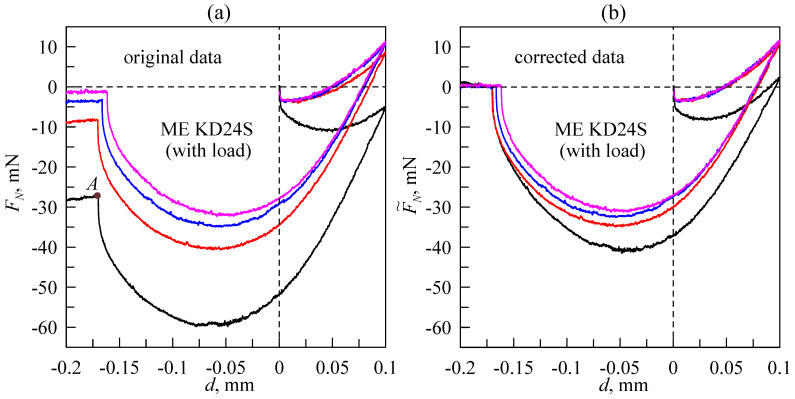
Dependencies of the normal force (*F_N_*) on the indentation depth (*d*) obtained during the indentation of a steel spherical indenter with a radius of *R* = 30 mm into a TARNAC CRG N0505 rubber substrate with a thickness of *h* = 5 mm. (**a**) The original data measured in four experiments performed under the same conditions. (**b**) The corrected dependencies after subtraction of the supposed signal drift. Curves, corresponding to the same experiments, are denoted by the same color in both panels.

**Figure 7 sensors-23-00456-f007:**
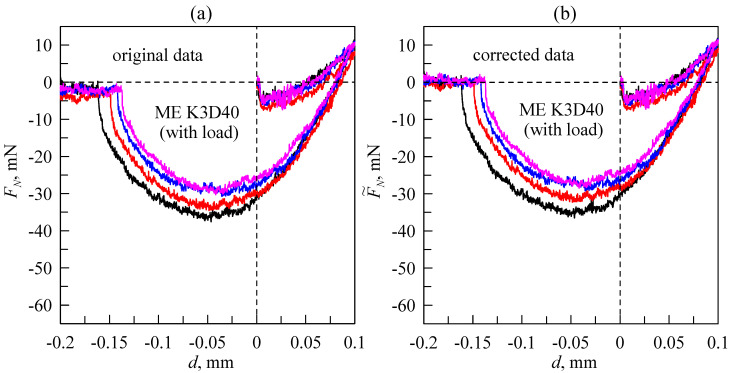
Dependencies of the normal force (*F_N_*) on the indentation depth (*d*), similar to curves shown in Figure 6 but measured by the K3D40 force sensor. Original (**a**) and corrected (**b**) data. Curves, corresponding to the same experiments, are denoted by the same color in both panels.

**Figure 8 sensors-23-00456-f008:**
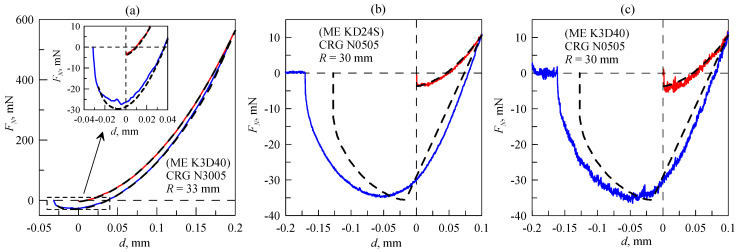
(**a**–**c**) Dependencies (*F_N_*(*d*)) obtained in experiments with the designed facility (solid lines) compared to numerical simulation (dashed lines) performed within the BEM model. Each panel shows a different experiment. Dependencies shown in panels (**b**,**c**) were measured under the same conditions with different force sensors.

**Figure 9 sensors-23-00456-f009:**
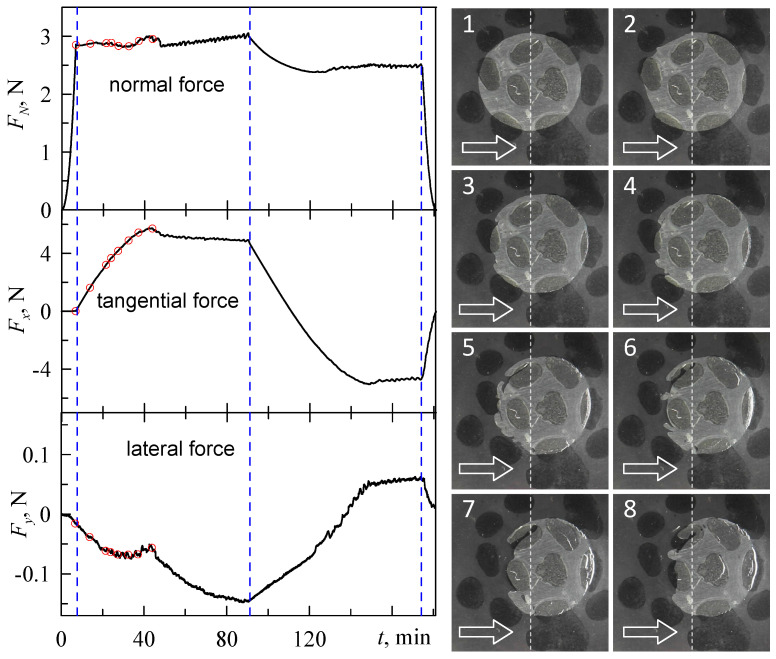
Time dependencies of all three components of the contact force ((**left**) panels) and related images of the contact area ((**right**) panels). Chemical inhomogeneities on the surface of the indenter in the (**right**) panel are pictured as dark spots. 8 red dots on the curves (**left** panel) show the time moments when the snapshots (**right** panel) were taken. Increasing numbers of the snapshots relate to growth of both tangential shift of the indenter and time from the start of the experiment.

## Data Availability

The datasets generated for this study are available upon request from the corresponding authors.

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
