# Peer review of "High-Precision Tribometer for Studies of Adhesive Contacts"

_sensors, 2023, doi:10.3390/s23010456_

Round 1

Reviewer 1 Report

Dear authors,

Your research is one interesting to find the best solutions to improve the accuracy of the indentation the soft materials and obtained information reefer to the adhesion process.

To improve the quality of the manuscript please clarify following questions :

1. In Fig. 1-c  is not clearly presented the indenter and the sensor. A detail with the indenter will be necessary.

2.In Fig. 2 The normal force is positive and in Fig. 5 the normal forces are negatives. Must be indicate if the normal force is positive or negative when the indenter  is pressed on sample. Also, is not clearly if indentation depth has positive or negative values in Fig.5. 

3. How can the adhesion be interpreted according to the  graphics from Fig. 5 and 6?

Author Response

  1. In Fig. 1-c is not clearly presented the indenter and the sensor. A detail with the indenter will be necessary.

Answer:

We have added the enlarged picture of the indenter and force sensor (please see Figure 2 in the revised version).

2. In Fig. 2 The normal force is positive and in Fig. 5 the normal forces are negatives. Must be indicate if the normal force is positive or negative when the indenter is pressed on sample. Also, is not clearly if indentation depth has positive or negative values in Fig.5.

Answer:

We introduced Section 3. Data Analysis and Interpretation where we added explanation of such type of behavior. Namely:  

“After the moment of the first contact, contact area is expanding (see Video 1), and normal force is negative F(= 0) < 0 N. Negative force at zero indentation depth d during indentation phase is clearly visible in Figure 8 (b) and (c). Dependencies in Figure 8 (a) are shown in wider range of a normal force so mentioned behavior is not observed, however it is clearly visible in the inset. Note that adhesive strength of the contact (maximal absolute value of the adhesive force during pull off at d < 0 mm) is approximately the same for all cases and equals FA ≈ 30 mN. Maximum of the adhesive force relates to the minimum of the F(d) dependence during the pull-off.”

  1. How can the adhesion be interpreted according to the graphics from Fig. 5 and 6?

Answer:

We added the detailed explanation of the performed experiments. Please see Section 3. Data Analysis and Interpretation, Eq. (2)-(6) and Figure 8 in the revised version of manuscript.

Reviewer 2 Report

The paper outlines the basic principles and scheme for developing new device for measuring adhesive forces with the ability to measure not only normal, but also tangential forces. The latter is the main element of novelty, according to the reviewer. The problems encountered by the authors in the design and use of the device are described in detail, which may be useful to other researchers and developers of devices. The relevance of the topic is related to the need to take into account adhesive forces during indentation and frictional contact, if the applied forces are sufficiently small. The paper may be published subject to the following remarks:

1. What is the sensitivity limit of the device? Good results have been obtained for rubber, a material with high adhesive and deformation properties. The paper

Myshkin, N.K., Goryacheva, I.G., Grigoriev, A.Y. et al. Contact Interaction in Precision Tribosystems. J. Frict. Wear 41, 191–197 (2020).

 shows a scheme of adhesion measuring device and shows the results for silicon, this is another scale level than rubber. It would be useful to make a comparison with this and other similar results and devices.

2. There are no examples of using a lateral force sensor, at the same time, the title contains the word Tribometer.

3. How qualified should the person using the device be?

4. In the list of references, 28 percent of self-citations, is this not too much? Usually, with rare exceptions, up to 20 percent is considered normal.

Author Response

  1. What is the sensitivity limit of the device? Good results have been obtained for rubber, a material with high adhesive and deformation properties. The paper

Myshkin, N.K., Goryacheva, I.G., Grigoriev, A.Y. et al. Contact Interaction in Precision Tribosystems. J. Frict. Wear 41, 191–197 (2020).

 shows a scheme of adhesion measuring device and shows the results for silicon, this is another scale level than rubber. It would be useful to make a comparison with this and other similar results and devices.

Answer:

We thank Referee for bringing this feature to our attention, we added related comments and reference (ref [52-55] in revised version) in the text (please see Section 5 Summary and discussion):

“It is worth to mention that sensitivity of the designed equipment is limited by the abilities of the installed force sensor, thus it can be increased by using the more capable sensors. Also, even though functional modification of the device for certain type of experiment is possible, more advanced studies or experiments in unique conditions require special type of tribometers. As a notable examples of such tribometers we can refer to millitribometer MTU-2k7 designed to study small-sized friction units and small loads [52], six-axis force/torque sensor for slow friction measurements under ultra-high vacuum and other conditions [53], setup for studying friction-induced vibrations [54] and haptics [55].”

  1. There are no examples of using a lateral force sensor, at the same time, the title contains the word Tribometer.

Answer:

We introduced Section 4. Investigation of the tangential contact in the revised manuscript where we described the tangential contact and presented obtained data. Also, Video 3 showing the tangential contact is also added in supplementary material.

  1. How qualified should the person using the device be?

Answer:

Since there are no any hazardous factors present in the device, no special training needed to operate the equipment, besides basic knowledge of physics and PC.

  1. In the list of references, 28 percent of self-citations, is this not too much? Usually, with rare exceptions, up to 20 percent is considered normal.

Answer: We thank Referee for bringing this feature to our attention. Most of the self-citation show the application of the designed device, therefore are important for article context. Thus, we have added more references in the revised manuscript and reduced self-citation.

Reviewer 3 Report

This research paper is concerned with High precision tribometer for studies of adhesive contacts. This manuscript is not well structured and not well written. It is more like a description of design for a setup and not an academic paper.

Author Response

We respectfully disagree with the Referee and we presumed that our paper can be useful for readers. Moreover, in the revised version we have added new sections concerning data interpretation and tangential contact which we believe can make our paper more academic.

Round 2

Reviewer 2 Report

The revised paper can be accepted. 

Reviewer 3 Report

This version is suitable for publishing.